# Discovery of Two New *Astyanax* Cavefish Localities Leads to Further Understanding of the Species Biogeography

**Luis Espinasa [1,*], Claudia Patricia Ornelas-García [2], Laurent Legendre [3], Sylvie Rétaux [4], Alexandra Best [1], Ramses Gamboa-Miranda [5], Hector Espinosa-Pérez [2] and Peter Sprouse [6]**

1 School of Science, Marist College, 3399 North Rd, Poughkeepsie, NY 12601, USA; Alexandra.Best1@marist.edu

2 Departamento de Zoología, Instituto de Biología, Universidad Nacional Autónoma de México, Tercer Circuito Exterior S/N. Mexico City CP 045110, Mexico; patricia.ornelas.g@ib.unam.mx (C.P.O.-G.); hector@ib.unam.mx (H.E.-P.)

3 UMR 9191 EGCE, CNRS, IRD, Université Paris-Saclay, 91198 Gif-sur-Yvette, France; Laurent.Legendre@egce.cnrs-gif.fr

4 Equipe Développement Evolution du Cerveau Antérieur, Paris-Saclay Institute of Neuroscience, CNRS and University Paris-Sud and Paris-Saclay, 91198 Gif-sur-Yvette, France; sylvie.retaux@inaf.cnrs-gif.fr

5 Instituto de Energías Renovables, Universidad Nacional Autónoma de México, Temixco, Morelos 62580, Mexico; ramiga@ier.unam.mx

6 Zara Environmental LLC, Manchaca, TX 78652, USA; Peter@zaraenvironmental.com

* Correspondence: luis.espinasa@marist.edu

**Abstract:** The *Astyanax* species complex has two morphs: a blind, depigmented morph which inhabits caves in México and an eyed, pigmented surface-dwelling morph. The eyed morph can also be found in a few caves, sometimes hybridizing with the cave morph. This species complex has arguably become the most prominent model system among cave organisms for the study of evolutionary development and genomics. Before this study, 32 caves were known to be inhabited by the cave morph, 30 of them within the El Abra region. The purpose of this study was to conduct new surveys of the area and to assess some unconfirmed reports of caves presumably inhabited by troglomorphic fish. We describe two new localities, Sótano del Toro #2 and Sótano de La Calera. These two caves comprise a single hydrologic system together with the previously described cave of Sótano del Toro. The system is inhabited by a mixed population of troglomorphic, epigeomorphic, and presumably hybrid fish. Furthermore, *Astyanax* cavefish and the mysid shrimp *Spelaeomysis quinterensis* show a phylogeographic convergence that supports the notion that the central Sierra de El Abra is a biogeographical region that has influenced the evolutionary history of its aquatic community across species. The presumptive location of the boundaries of this biogeographical region are identified.

**Keywords:** *Astyanax*; *Spelaeomysis*; *Troglomexicanus*; *Speocirolana*; Toro cave; Sierra de El Abra; troglomorphy; troglobite; stygobite

## 1. Introduction

Cave organisms have been a choice model for many evolutionary biologist due to their independent loss of vision and pigmentation. The genetic and developmental controls on troglomorphic features are best understood in the blind Mexican tetra [1]. While most recent authors consider it to be the cave morph of *A. mexicanus* De Filippi 1853, some authors gave it the status of species, *Astyanax jordani* Hubbs & Innes 1936. Most notably, *A. jordani* is still used in the IUCN Red List and listed as Vulnerable [2].

Both surface and cave morphs remain inter-fertile, making the species complex well-suited for experimental manipulations [3]. With the genomes of both morphs already sequenced, *Astyanax* is one of the main contributors in the understanding of cave evolution, and is also recognized as an influential model system in evolutionary developmental biology [4,5].

Mitchell et al. [6] described 29 cave localities for troglomorphic *Astyanax*, all of which occur within the El Abra region (which includes the adjoining Sierra de El Abra, Sierra de Guatemala, and Micos area), in the states of San Luis Potosí and Tamaulipas in Northern México. Two additional cave localities, Granadas and La Joya [7,8], have been described for the closely related species of *Astyanax aeneus* Günther, 1860 in Guerrero, Southern México. The last and most recently discovered locality was Cueva Chiquitita [8], from Sierra de El Abra (Figure 1).

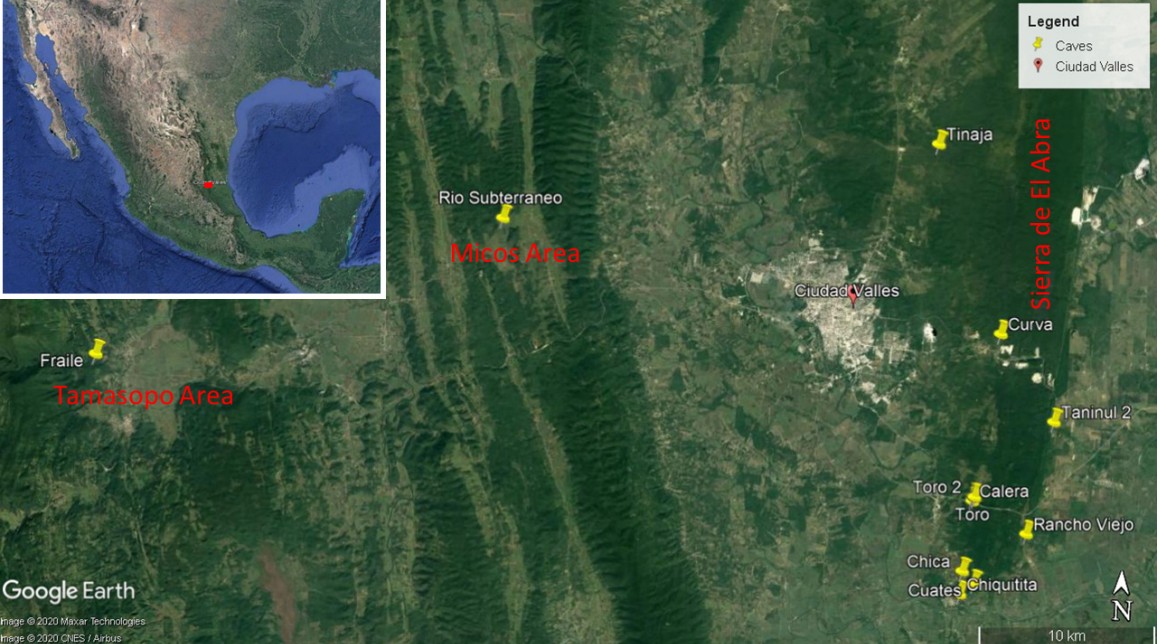

**Figure 1.** Cave locations. Tinaja and Curva caves are in the central Sierra de El Abra. Taninul, Calera, Toro, Toro #2, Rancho Viejo, Chica, Chiquitita and Cuates are in southern Sierra de El Abra. Río Subterráneo is in the Micos area, still within the El Abra general region. Cueva del Fraile is in the Tamasopo area. Pachón Cave is outside the map boundary, 70 km north of Curva, in the northern Sierra de El Abra. Molino and Caballo Moro are about 110 km north of Curva, in the Sierra de Guatemala. Map showing location of Ciudad Valles, San Luis Potosi. Google Earth, earth.google.com/web/.

Despite the presence of thousands of caves, there are no known troglomorphic *A. mexicanus* ssp. *jordani* sites south of the Río Tampaón. Cueva Chiquitita is a spring found at the southernmost edge of the Sierra de El Abra, approximately 200 m from the Río Tampaón. Cueva Chiquitita is inhabited by surface, troglomorphic, and hybrid fish. It is also likely that there is a continuous underwater passage between Cueva Chiquitita and one of the most cited *Astyanax* cave localities, Cueva Chica [6,8]. Subsurface hydrologic connections allow for both troglomorphic and surface fish to migrate between El Abra cave localities. These connectivity is supported by gene flow [9] and geologic [10] studies.

Previous reviews of caves inhabited by troglomorphic *Astyanax* suggested areas that could yield new discoveries, in particular the southern Sierra de El Abra [11]. This is a low-lying area that has little potential for deep caves. Most caves were explored in the 1970's, and cavers at that time may have put less effort into describing shallow caves that would nevertheless be of biological interest [6,8]. Supporting evidence for the possibility of new discoveries was the aforementioned recent discovery of Cueva Chiquitita in this area [8].

Previous mitochondrial DNA analyses on *Astyanax* cavefish have shown the presence of two broadly different clades that have been termed mitochondrial lineages A and B [12]. Caves harboring populations with haplotype B mitochondrial DNA (Sabinos, Tinaja, Piedras and Curva) are restricted to an area in the central Sierra de El Abra [13]. El Abra caves are also inhabited by the mysid cave shrimp *Spelaeomysis quinterensis* Villalobos, 1951. Histone H3 DNA sequences of them also showed that mysid shrimp from those caves in the central Sierra de El Abra were quite different from other cave populations [13]. This phylogeographical convergence supports the hypothesis that the central Sierra de El Abra is a biogeographical zone, perhaps with barriers for either cave-to-cave or surface-to-cave gene flow, that have modulated the evolutionary history across species in the aquatic community. Interestingly, Sótano del Toro is located between the central and the southern Sierra de El Abra. Determining which stygobitic community it belongs to will allow for a better understanding of where the biogeographic barriers are located.

Another area that was identified [11] that could yield interesting new discoveries was the Tamasopo area and, in particular, Cueva del Fraile. This cave is 24 km west of the nearest *Astyanax* cave population in Río Subterraneo, in the Micos area [11]. This locality is in a drainage isolated from the El Abra region hydrology by the 100 m waterfall of Tamul. If there is a cave-adapted population in the Tamasopo area, it would have to be from an independent evolutionary event, and thus of great scientific interest.

## 2. Materials and Methods

Sótano del Toro and Sótano del Toro #2 were visited on March, 2019 and on January 2020. Sótano de La Calera, near Sótano del Toro, was explored in June 2019 and in January 2020. Cueva del Fraile was visited in February and shortly after in March 2019. Cueva del Rancho Viejo, Taninul #2, and Tinaja were visited in January 2020. Figure 1 shows a map with the caves described in this article. Description of the caves and their precise ubications can be found in [6,11].

To explore Sótano del Toro #2, Sótano de La Calera, Taninul #2, and Tinaja, single rope techniques were employed to ascend and descend their up to 30 m pits. Some pools received special attention with the help of mask, snorkel and underwater scuba lights. Sótano de La Calera and Sótano del Toro #2 were surveyed using a DistoX2, and in the case of Toro #2 using the TopoDroid app on a tablet. Survey data were processed using Compass and Walls, and drawn using Adobe Illustrator.

To capture specimens, baited traps or hand-held nets were used. Photographs were taken in the field with the help of a small glass tank and a Canon EOS100 camera. A small fin clip was taken with scissors for DNA studies from all specimens. Voucher samples were stored at the Colección Nacional de Peces, IBUNAM, Mexico.

In accordance with the Ethical Guidelines for the Use of Animals in Research and collecting permit # SGPA/DGVS/1893/19 from the Secretaría del Medio Ambiente y Recursos Naturales, México issued to Patricia Ornelas-García, only five specimens were euthanized from Sótano del Toro #2, which were stored in 100% ethanol. Five other fish were kept alive in the laboratory of Patricia Ornelas-García for breeding in captivity, which can serve as a stock for future studies. All other specimens collected were returned unharmed to their source pool.

*Astyanax* mitochondrial 16S rRNA was sequenced from 5 individuals collected in Sótano del Toro #2. For comparison, DNA sequences were aligned to sequences from localities with reported [12,14] haplotypes of mitochondrial lineage A (2 Comandante surface river, 2 Sótano del Molino, 2 Sótano del Caballo Moro, 2 Cueva del Pachón, 1 Cueva Chica, and 4 Cueva Chiquitita) and from localities with haplotypes of mitochondrial lineage B (4 Rascón surface stream, 2 Tamasopo surface stream, 2 Cueva de Los Sabinos, 4 Cueva de la Tinaja, and 1 Cueva de la Curva). For mysid shrimps, the Histone H3 was sequenced from a single individual from Toro #2. It was then compared with sequences from localities with reported [13] haplotypes of clade A (2 Pachón and 1 Cueva Chiquitita) and from haplotypes of clade B (3 Tinaja and 1 Sabinos caves). The use of two different markers for comparison between *Astyanax* and mysid shrimp aligns with previous studies [13].

Genomic DNA samples were obtained following standard methods for DNA purification using Qiagen's DNeasy® Tissue Kit by digesting a fin clip of the individual in the lysis buffer. Markers were amplified and sequenced as a single fragment using the 16Sar (CGCCTGTTTATCAAAAACAT) and 16Sb (CTCCGGTTTGAACTCAGATCA) primer pair for 16S rRNA [15] and the H3aF (5' ATGGCTCGT ACCAAGCAGACVGC 3') and H3aR (5' ATATCCTTRGGCATRATRGTGAC 3') for histone H3 [16]. Amplification was carried out in a 50 µL volume reaction, with QIAGEN Multiplex PCR Kit. The PCR program consisted of an initial denaturing step at 94 °C for 60 s, 35 amplification cycles (94 °C for 15 s, 49 °C for 15 s, 72 °C for 15 s) and a final step at 72 °C for 6 min in a GeneAmp® PCR System 9700 (Perkin Elmer). PCR amplified samples were purified with the QIAquick PCR purification kit and directly sequenced by SeqWright Genomic Services. Chromatograms obtained from the automated sequencer were read and contigs made using the sequence editing software Sequencher™ 3.0. All external primers were excluded from the analyses. BLAST was used to identify GenBank sequences that resemble the specimens. Sequences were aligned with ClustalW2.

## 3. Results and Discussions

### *3.1. Southern Sierra de El Abra*

3.1.1. Background on Sótano del Toro, Cueva del Rancho Viejo and Taninul #2

Sótano del Toro is the smallest and shallowest of the known eyeless *Astyanax* caves. It is possible to stand in the entrance sink and see the 0.5 m wide by 2.5 m long pool. A gallery can be seen continuing underwater. Light reaches this small pool, as can be seen in Figure 2. The cave was visited in 1969, when three days of work resulted in only about eight fish being collected. On March 2008, three cavefish were obtained for genomic work [9]. Previous reports [11] conclude that "None of these fishes appeared to have hybridized with surface *Astyanax*, even though Cueva Chica (which has large amounts of surface and hybrids) is located only 4 km to the south". They further report that "Another cave may exist in the vicinity, but it was not found in 1974 by local guides who knew of it".

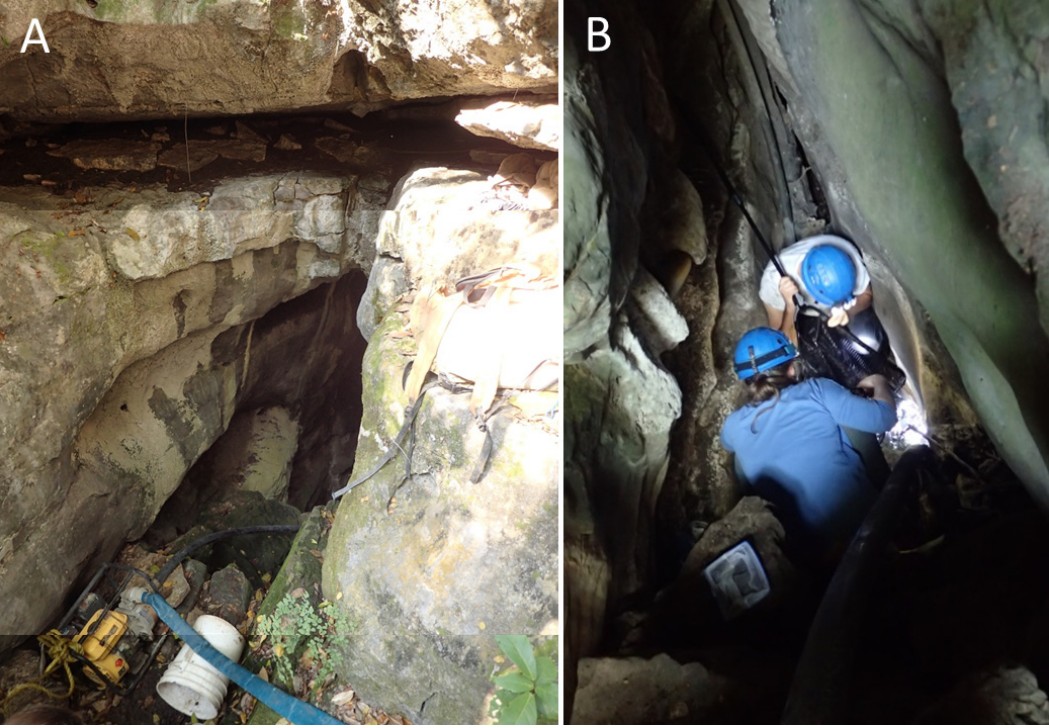

**Figure 2.** Sótano del Toro. (**A**); Entrance sink descends to a 0.5 m wide by 2.5 m long pool. (**B**); Light reaches this small pool. Locals use a water pump (A) to extract water for the local farm.

The Cueva del Rancho Viejo is near a small resurgence called Nacimiento del Rancho Viejo, which only discharges water after a rain. The cave was explored by Elliott et al. in 1974. A climbdown of 8 m reaches the first pool, 5-10 m deep. A low gallery 100 m long reaches a second pool. Eyed *Astyanax* and cichlids were observed and collected at that time [11]. During the dry season, no water comes out of the resurgence, and fish are isolated in pools that lie in complete darkness. This resurgence is of interest because it is only 3.6 km east of Sótano del Toro, and 4.2 km northeast of Cueva Chica, so they could be hydrologically related. Before this study it was unclear if troglomorphic fish were also present in this cave.

Cueva de Taninul #2 is 6 km northeast of Sótano del Toro. This cave is only 100 m away from the fresh water, non-sulphur resurgence pool at the hotel Taninul. This cave is about 200 m long, and has a 30 m pit. At the bottom of this pit there is a sump. This sump is at about the same level as the resurgence. There were no published records of fish inhabiting this cave before this study.

### 3.1.2. Results from Taninul #2, Cueva del Rancho Viejo, and Toro Area

Taninul #2 was explored to the final sump, where half a dozen fully eyed and pigmented fish were observed. They were comparable to surface fish found above ground, and were responsive to light. No troglomorphic fish were seen, despite careful observations while snorkelling with an underwater lamp.

We visited Cueva del Rancho Viejo on 6 January 2020. About 10 fully pigmented fish were observed in the pools. A baited trap was left with the aim of attracting troglomorphic fish, since they have enhanced smell [1,17]. The trap was then recovered on the 10th of January, 2020. Around 15 individuals were collected, but all had normal eyes and pigment. Fish were released unharmed after careful observation. Despite two days of effort and a baited trap, we found no evidence that troglomorphic or hybrid fish inhabit these caves. This supports the 1974 observation that only epigeomorphic fish inhabit this cave [11].

Exploration of the area around Sótano del Toro yielded two new entrances to caves with troglomorphic *Astyanax*: Sótano del Toro #2 is only 68 m west of Sótano del Toro, and Sótano de La Calera is 239 m south of Sótano del Toro #2.

Sótano del Toro #2 consists of a vertical pit of about 12 m. For the first 6 m the pit is very narrow, just 1 m wide. Then it opens into the roof of a large chamber that is 20 m long. In this chamber there is a 10 m diameter pool with fish (Figures 3 and 4). This pool is a sump heading in the general direction of Sótano del Toro. From the lake room several short dry galleries bifurcate, one of them being a 30 m crawl that reaches a low ceiling gallery with water, also inhabited by fish (Figure 5). This water passage heads in a southerly direction for another 20 m until a sump is reached, at which point it appears to continue in a direction that would suggest a possible connection with Sótano de La Calera. It is noteworthy that the walls near the sump are covered with fine mud inhabited by scores of nicoletiid insects, *Anelpistina quinterensis* (Paclt, 1979).

Sótano de La Calera has a 6 m pit entrance (Figure 6A). During the dry season when the cave was explored, galleries were interspersed with many pools varying from just 0.5 m long to over 10 m, some inhabited by *Astyanax* fish. Apart from the *Astyanax* and the nicoletiid insects, this cave system is inhabited by the mysid shrimp *Spelaeomysis quinterensis*, as well as *Speocirolana* isopods. The downstream gallery of Sótano de La Calera heads south for about 77 m via low ceiling crawlways until a sump is reached. This sump has not been explored by divers, and could lead to a significant extension of the system to the south. The upstream gallery has been mapped for 186 m and heads north, in the direction of Sótano del Toro #2 (Figure 7). Beyond the mapped limit, another 15 m have been explored until a crawlway is reached that is covered with fine mud and inhabited by scores of nicoletiids. There are also several unexplored side passages in Calera.

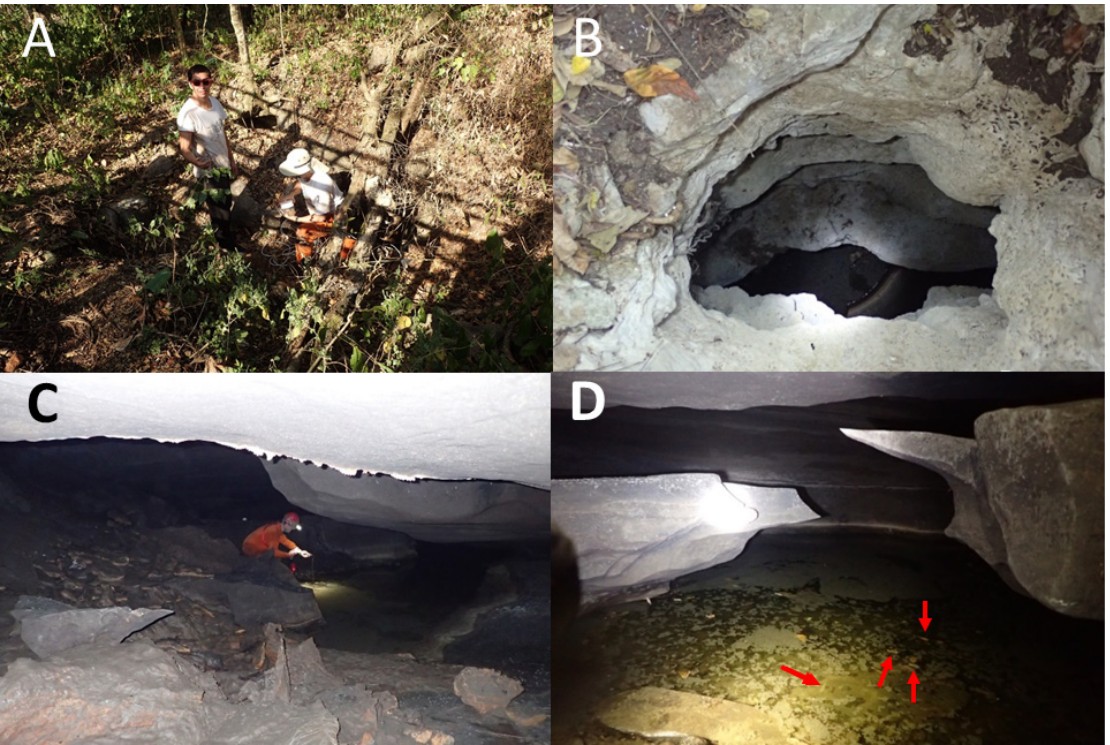

**Figure 3.** Sótano del Toro #2. (**A**); Entrance pit. (**B**); Narrow portion of the entrance pit. (**C**); Typical galleries within the cave. (**D**); Entrance lake. Arrows point to fish.

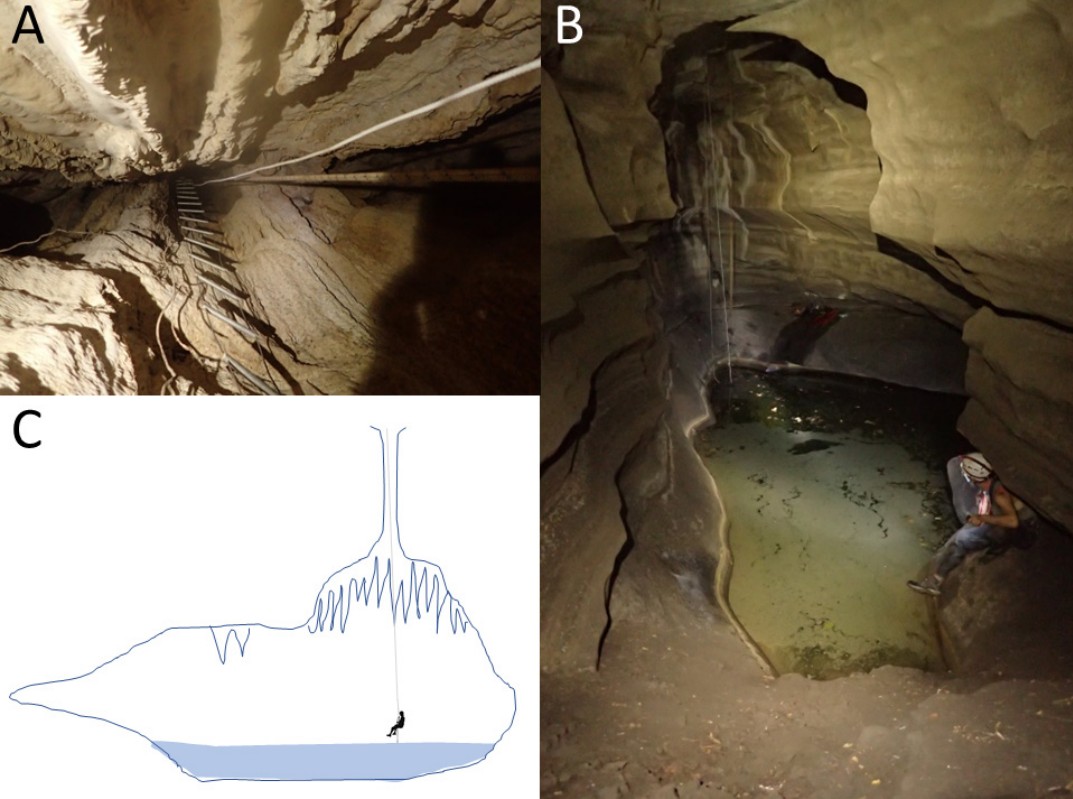

**Figure 4.** Sótano del Toro #2. (**A**); looking up to the narrow portion of the entrance pit. (**B**); Entrance lake. (**C**); Profile drawing of the entrance pit and lake.

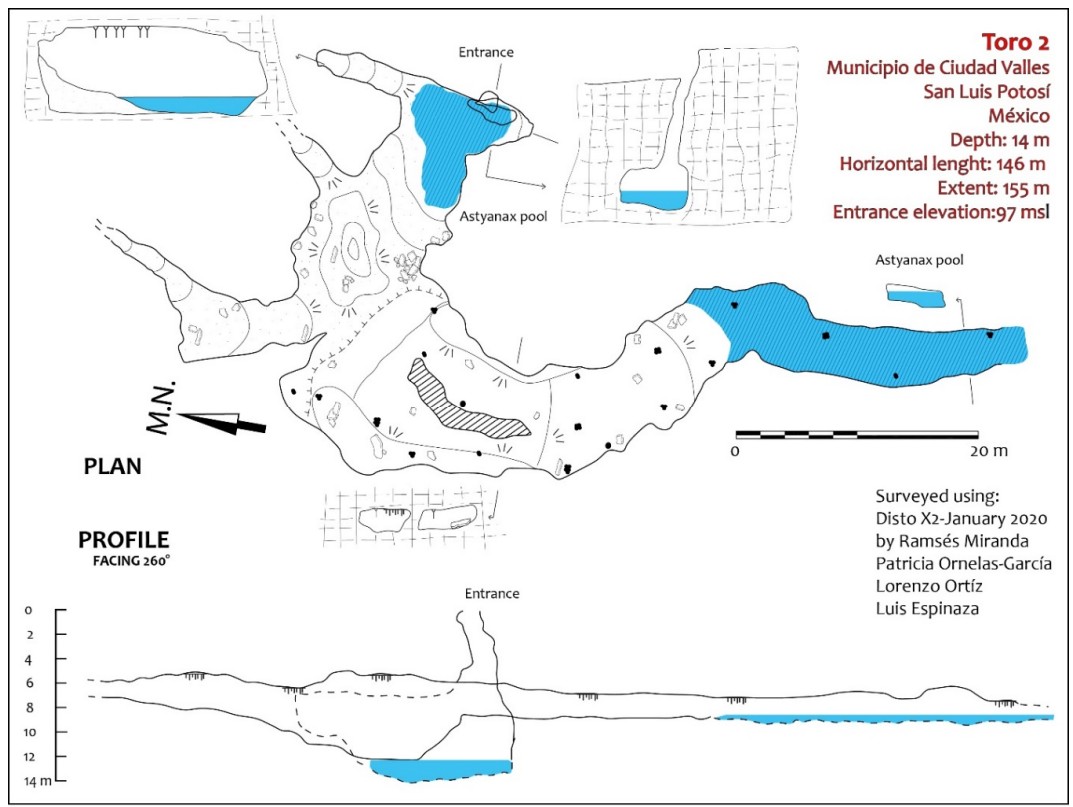

**Figure 5.** Map of Sótano del Toro # 2.

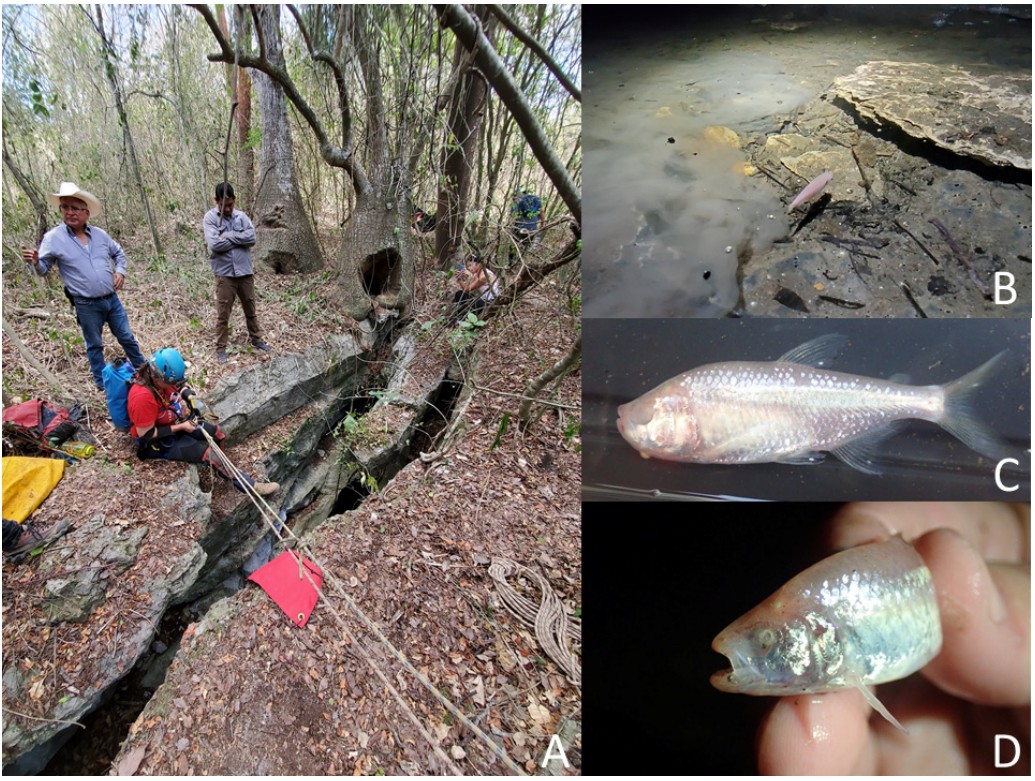

**Figure 6.** Sótano de La Calera. (**A**); Entrance pit. (**B**); One of the many pools with fish in this cave. (**C**); Fully troglomorphic fish. (**D**); Potential hybrid between troglomorphic and epigeomorphic fish. Notice reduced and embedded eyes.

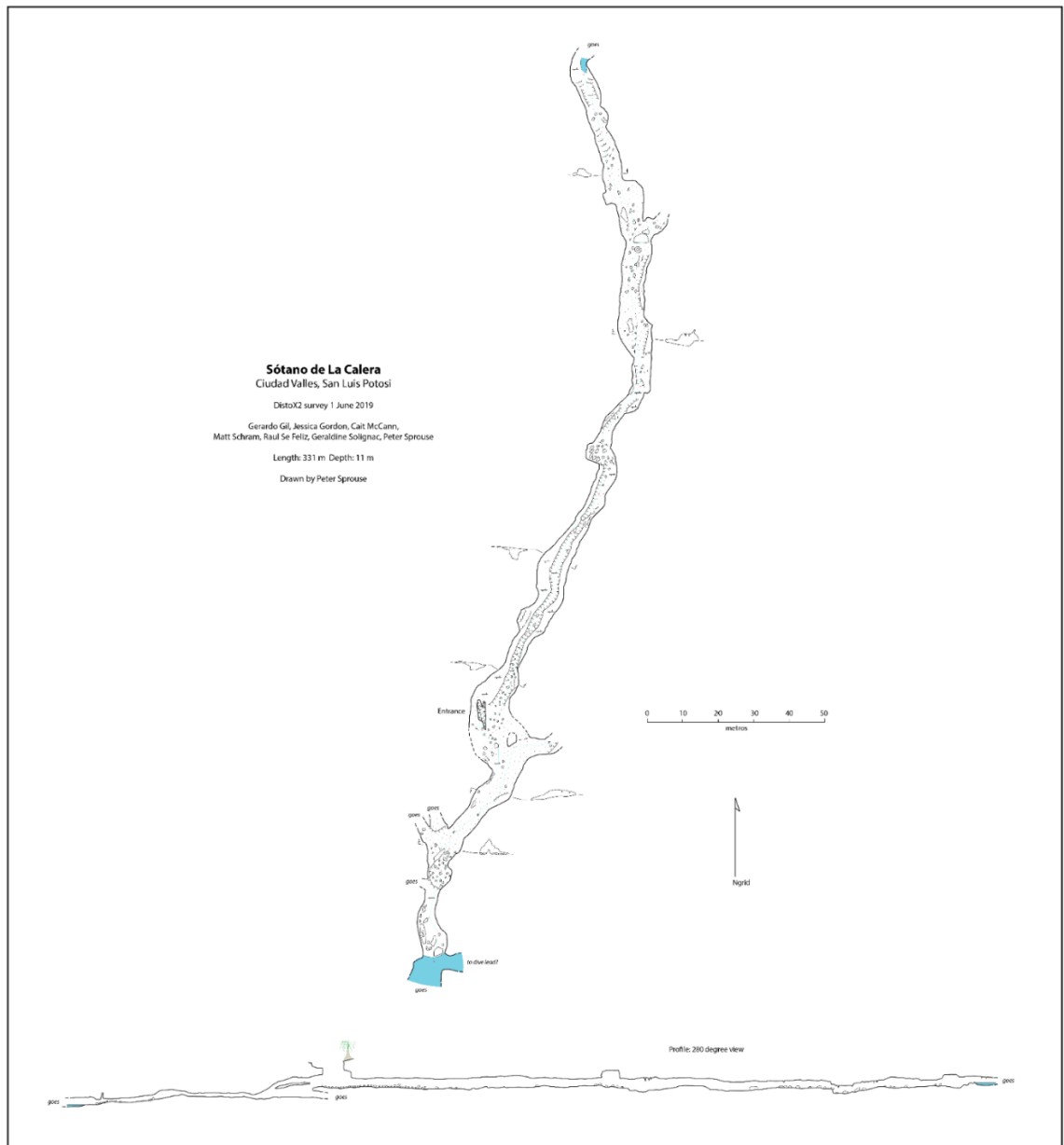

**Figure 7.** Map of Sótano de La Calera. During the dry season, galleries contain many pools, some inhabited by *Astyanax* fish. During the rainy season it becomes a flowing stream and fish can freely swim along its galleries and to Sótano del Toro and Sótano del Toro #2

The entrances to Sótano del Toro #2 and Sótano de La Calera are only 239 m from each other. Considering that Calera extends north for about 200 m, heading directly towards Sótano del Toro #2, and that Sótano del Toro #2 extends south for about 40 m south, it is likely that the muddy sections inhabited by nicoletiids are the opposite sides of a short sump. We suggest that both caves are a single cave system. These two caves also may be connected to Sótano del Toro, only 68 m away. It probably is a single hydrologic system with three entrances (Figure 8). During the rainy season, fish can probably travel easily from one section to the other.

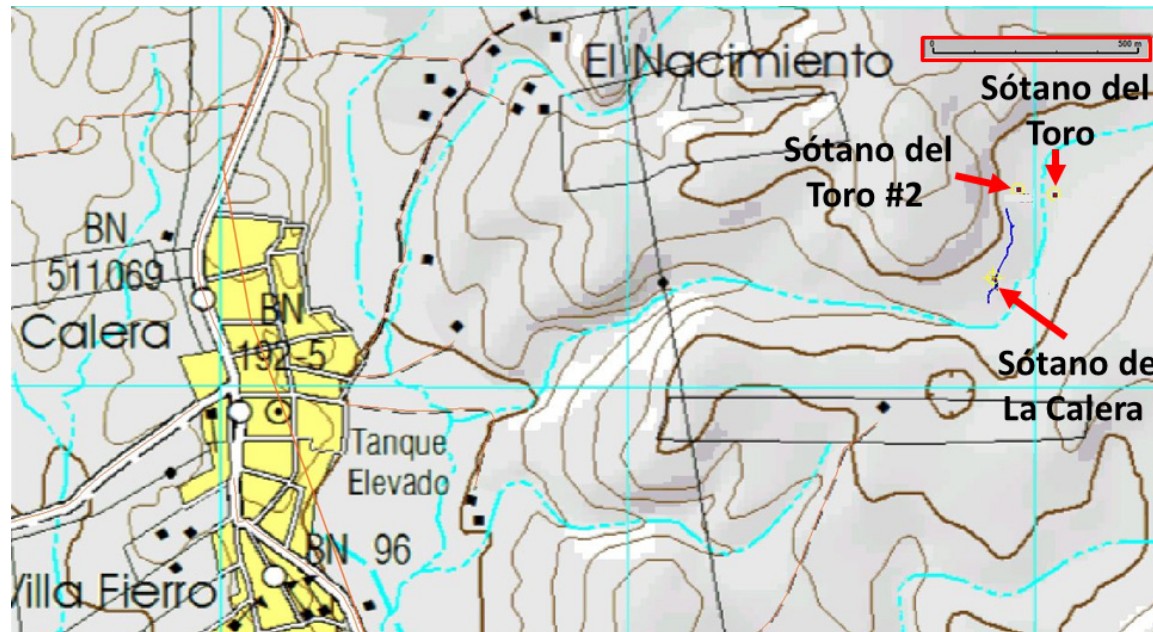

**Figure 8.** Topographic map showing the close proximity of Sótano del Toro, Sótano del Toro #2 and Sótano de La Calera. Also shown in the map are the town of La Calera, from which the cave receives its name, and the town of El Nacimiento, which means "spring". This spring may be one of the potential sources of surface fish found in the cave system. In dark blue is the line plot of Sótano de La Calera. Notice that it is heading directly towards Sótano del Toro #2. While sumps prevent passage without diving gear, these three caves likely form a single hydrologic system, and fish can move from one to the other during the rainy season. Modified map from INEGI data, F14D11, El Pujal, 1:50,000 topographic sheet.

From the point of view of the *Astyanax* biology in the Toro system, our observations differed significantly from previous reports [11] of a cave inhabited exclusively by troglomorphic fish. Instead, we found that the population is a mixed population (Figure 9). Some were highly depigmented, with characteristic pinkish-white coloration and no external eyes. Other individuals were identical to surface fish, fully pigmented, with large eyes, and responsive to light. But many individuals had high variability in the eyes, with individuals having phenotypes such as small eye size (Figure 9), closed pupil, embedded eye (Figure 6D), and eye mostly absent. Introgression between the surface morph and the cave morph is suggested by the presence of individuals that are highly depigmented, but with eyes (Figure 9G,H,K) or individuals with pigment, but reduced eyes (Figure 9J).

None of these three caves has a large bat population. No large streams enter them. While the entrances may provide debris that can serve as food for the fish, large amounts of food are likely only in the pools directly under the entrances, and not throughout the cave system. Troglomorphic, epigeomorphic and presumed hybrids are found throughout the caves. Future studies may resolve if there are significant different ecological pressures in the different pools to promote intraspecific variability, although since during the rainy season fish from all pools are intermixed, we doubt it will have a large impact in the overall population.

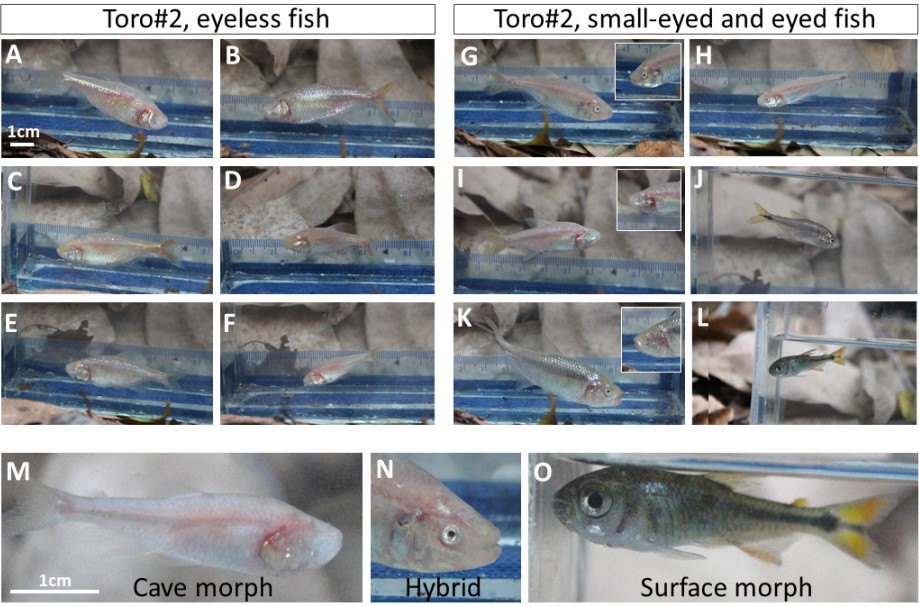

**Figure 9.** High variability in the eye and pigmentation level within the population inhabiting the Toro system. (**A–F**); Eyeless fish. (**G–L**); Small eyed and large eyed fish. (**M–O**); Higher magnification of representative morphs. Introgression between the surface morph and the cave morph is suggested by the presence of individuals that are highly depigmented, but with eyes (**G,H,K**) or individuals with pigment, but reduced eyes (**J**). Asymmetry is evident in some individuals (**K**) in which on one side they may have an eye, but on the other is eyeless (inset photo). Scale bar in A is for (**A–L**) and scale bar in M is for (**A–O**).

Five troglomorphic fish from the Toro system were genotyped. DNA amplification of mitochondrial 16S rRNA produced a 572 bp sequence. All specimens presented a haplotype A (GenBank# AP011982.1) identical in sequence to fish from Pachón, Chica, and Chiquitita caves and from the local surface *Astyanax* (Figure 10). When compared to the mitochondrial haplotypes B of fish from Rascón and Tamasopo surface streams, Toro specimens differed by 2–3 bp. When compared to the mitochondrial haplotypes B of cavefish from Sabinos, Tinaja, and Curva, they differed by 5 bp (red arrows in Figure 10). It is thus supported that *Astyanax* Toro specimens have a mitochondrial DNA most similar to haplotypes A, with specimens being identical to individuals from other southern El Abra caves like Chica and Chiquitita.

In the mysid shrimp from the Toro system, the H3 fragment was 328 bp long. Two clades or haplotypes (GenBank # MH422492, MH422494) were found, in support of Kopp et al. [13]. The two mysid shrimp haplotypes differed by 36 bp (10.9%). The first clade A included specimens from Pachón, Toro #2, and Chiquitita. The clade B included Tinaja and Sabinos. Thus, similarity of sequences among populations did not follow geographical proximity between caves. Specimens from the northernmost (Pachón) and the southernmost (Toro #2 and Chiquitita) portions of the Sierra de El Abra were identical. Likewise, the specimens from the central Sierra de El Abra (Sabinos and Tinaja) were identical. It is thus supported that the mysid shrimp from Toro #2 cave has an H3 DNA most closely related to the clade A, with their sequence being identical to individuals from other southern El Abra caves such as Cueva Chiquitita.

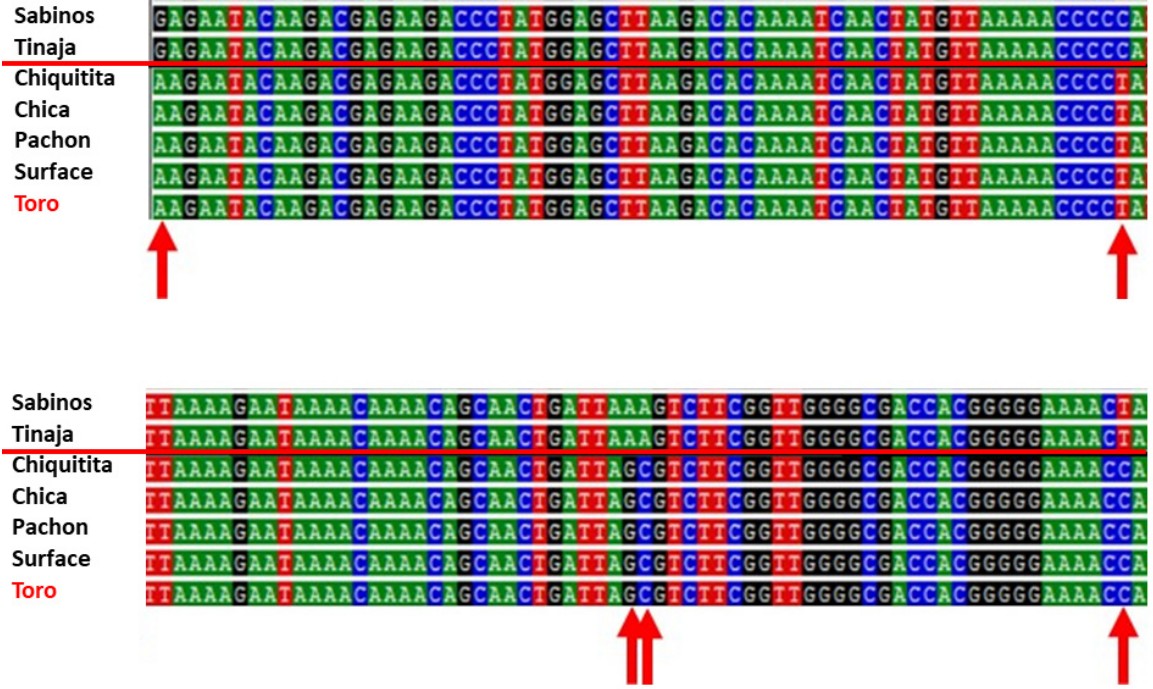

**Figure 10.** Fragment of the *Astyanax* mitochondrial 16S rRNA. Individuals from Toro system have identical sequence to populations harboring haplotype A (Cueva Chiquitita, Cueva Chica, Pachón caves, and Río Comandante surface river). Populations harboring haplotype B (Sabinos and Tinaja caves) have five bp disagreements in this fragment, indicated by red arrows.

### 3.1.3. Discussion: Southern Sierra de El Abra

Our results revealed some interesting aspects of the cave populations in the southern Sierra de El Abra. Previous reports have described Sótano del Toro as a small cave with a population composed of only troglomorphic *Astyanax*. Instead, we have found this to be a cave system with over 350 m of passages, three different entrances, and many environmentally variable pools inhabited by a mixed population of troglomorphic, epigeomorphic and presumably hybrid fish, similar to those found in Chica, Cuates and Chiquitita caves. Toro system cavefish also have the mitochondrial A haplotype found in the aforementioned southern Sierra de El Abra cave populations, and in surface fish from the streams that surround Sierra de El Abra. They are distinct from the cavefish populations of the central Sierra de El Abra, and the surface fish from Rascón.

What is the origin of the surface fish in the Toro system? The cave is heading directly south, towards Chica, Cuates, and the resurgence of Chiquitita. The Chiquitita spring has been hypothesized to be the source and entrance point of surface fish to these caves [8]. The Chiquitita resurgence is about 5.6 km to the south. Another alternative is that the cave system changes its direction and veers to the west to a spring near the town of "Ojo de Agua", about 1.3 km away. Regardless of the surface fish's source, the Toro system shows that they have the potential to penetrate deep into the cave systems. These surface fish are swimming upstream starting at a spring. They are not washed in. Furthermore, surface fish were found in galleries in complete darkness late in the dry season, so they must have been there for many months. Their presence in large numbers in a cave far from any surface stream input implies that, to a certain extent, they can survive and reproduce alongside the cave morph. While they may be slimmer than troglomorphic fish, they are somewhat successful in reproducing, as evidenced by the high number of apparent hybrids observed.

It has been argued that surface fish at Cueva Chica have been able to survive and successfully reproduce because of the mild selective conditions due to an ample food supply from the large bat

colony in this cave [6]. The Toro system shows that even without a large input of food, surface fish can still be part of the reproductive community over the span of six months or more. This contrasts drastically with the view that surface fish are to be considered accidentals, destined to die soon after, such as we have personally witnessed in the Río Subterráneo cave in the Micos area. This result is corroborated by the Cuates cave population, also in the southern El Abra, where fish of both morphs cohabitate.

A recent study has estimated that all cavefish populations are probably recent, less than 20,000 years old [18]. It has been argued that the incongruence of the mitochondrial DNA phylogeny with phylogenies obtained with several independent nuclear loci does not support the existence of two cavefish lineages [18]. Genomic studies have further shown a much more complex evolutionary history with reticulation, with recent and historical gene flow both within and between cave and surface populations where "a simple bifurcating tree does not fully capture the history of these populations" [19]. Regardless of whether or not there are cave-adapted fish lineages, results show that caves harboring populations with haplotype B mitochondrial DNA (Sabinos, Tinaja, Piedras and Curva) are restricted to a distinct area in the central Sierra de El Abra [13]. The Toro system is now recognized as belonging to the southern Sierra de El Abra, which harbors *Astyanax* mitochondrial haplotype A. A consistent pattern was corroborated with the stygobitic mysid shrimp, also suggesting that those in the central Sierra de El Abra (Sabinos and Tinaja) harbor a separate haplotype different from the rest of the Sierra de El Abra populations (Pachón, Toro #2, and Cueva Chiquitita).

Phylogeographic results obtained from the mysid Shrimp, *Spelaeomysis quinterensis*, are consistent with the results of mitochondrial studies in *Astyanax*. This suggests that the geographic distribution of mitochondrial haplotypes in *Astyanax* is not stochastic. It has been suggested that northern populations like Pachón, Molino, and Caballo Moro and the southern Sierra de El Abra populations have haplotype A, while the central populations have haplotype B because they represent different colorizations and/or because they have been affected by recent introgression from surface populations [20]. It has also been suggested that the disparity between nuclear and mitochondrial DNA could be explained by linkage to paternally or maternally inherited adaptive characters of distinct populations [20]. The present data also support that the Sierra de El Abra has distinct biogeographic areas, with partial barriers that affect evolutionary histories and generate evolutionarily significant units across different species of the aquatic cave community. Since the Toro system is the northernmost cave of the southern Sierra de El Abra group and Curva cave is the southern-most cave of the central group, it is now recognized that the border between these two groups is probably between 21°58′ N and 21°54′ N.

We propose a model to explain the biogeography of the Sierra de El Abra, in which its low-lying southern portion is like a sponge or sieve where conduits have few barriers for surface fish entering at resurgences along the Río Tampaón (28.5 m above sea level masl) or into the Río Valles (48 masl). Chiquitita's bottom is at 27 masl., Chica at 30, Cuates at 37, and Toro at 88 [11]. All these caves have mixed populations with epigeomorphic fish and troglomorphic fish with mitochondrial A haplotypes. In the central Sierra de El Abra caves, populations are highly troglomorphic, with little incidence of hybridization, and harboring mitochondrial B haplotypes. Kopp et al. [13] have proposed a biogeographic divide that isolates both *Astyanax* cavefish and mysid cave shrimps in the central Sierra de El Abra caves from those in the southern Sierra de El Abra. The southernmost cave population of the central Sierra de El Abra group is Curva, whose bottom is at 113 masl. The central and southern portions of the Sierra de El Abra are separated by distance and altitude barriers, plus there is the remnant of a relict canyon where the Río Tampaon/Valles crossed farther north before plate tectonics elevated the southern portion [10]. This relict canyon is used by the highway that connects Ciudad Valles with Tamuin (Figure 11A). It has been estimated that the Río Tampaon/Valles changed its northern course to its current locality about 2.78–0.69 mya [10]. The authors argued that the southern portion of El Abra was elevated, and cave erosion could not start until after that time [10]. Caves north of this southern, low-lying sponge may effectively be isolated from surface fish immigrants, with the

exception of Sótano de Yerbaniz and Sótano de Matapalma, where surface fish are flushed into entrance pits [11].

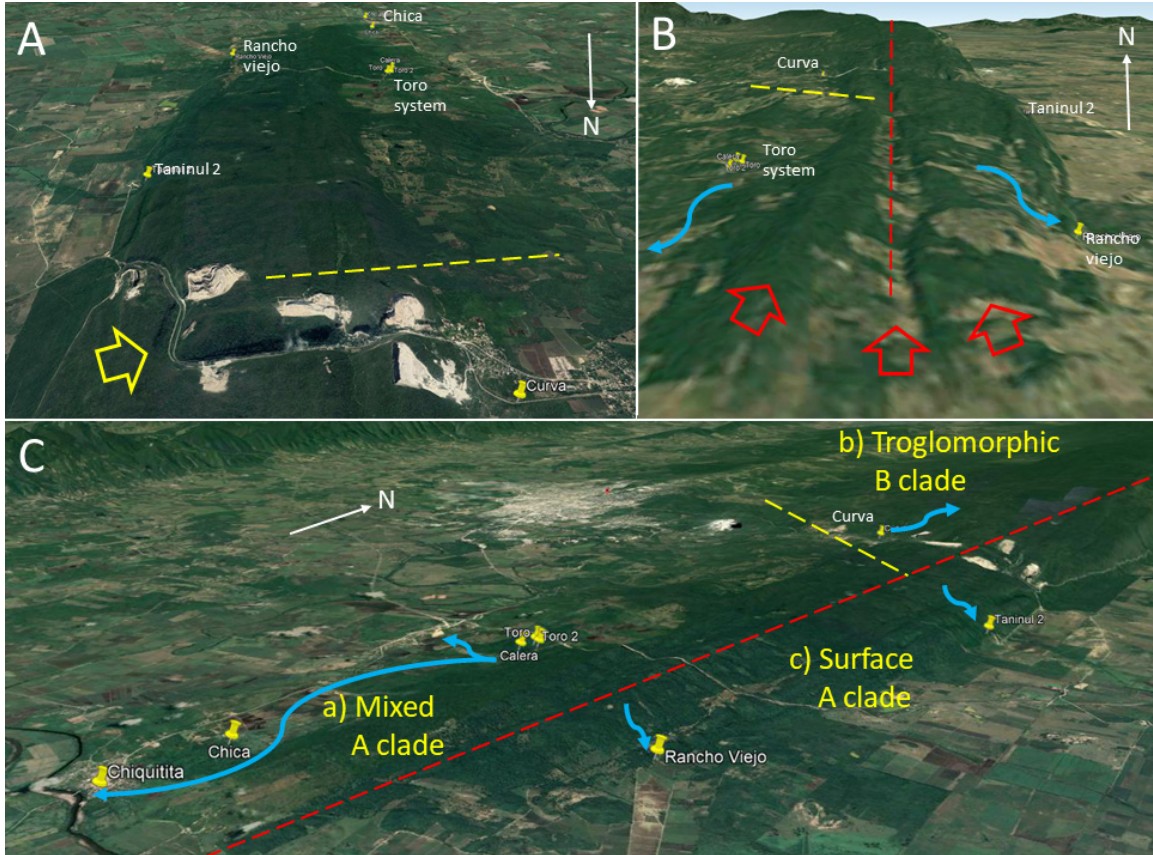

**Figure 11.** Model of biogeographical barriers (dashed lines). (**A**); There is a difference in altitude (Toro at 88 masl and Curva at 132 masl) and a fossil canyon (Yellow arrow) that divides the central and low-lying Southern sections of the Sierra de El Abra (**B**); The southern El Abra ridge is made of two crests and a valley (Red arrows). It is proposed that these tectonic structures may have promoted cave development resulting in two watersheds (blue arrows). (**C**); The combination of these two biogeographic barriers has prompted the development of three differently isolated communities: (a) In the low-lying southwest, *Astyanax* populations are mixed, and both cavefish and mysid shrimp harbor haplotype A markers; (b) To the north of the canyon, *Astyanax* populations are mainly troglomorphic, with little hybridization with surface fish, and both cavefish and mysids harbor haplotype B; (c) To the east, only surface fish are encountered in the caves. Map showing location of Ciudad Valles, San Luis Potosi. Google Earth, earth.google.com/web/.

A restriction to this low-lying sponge model of easy fish dispersal lies to the east. Both Cueva del Rancho Viejo and Taninul # 2 have a bottom at about 50 masl. Neither of them has evidence of cavefish. Only surface fish that derived from the nearby local surface streams inhabit them. Barriers on the east flank of the Sierra de El Abra for *Astyanax* cavefish dispersal seem to be prevalent throughout the range. None of the caves or resurgences on the eastern side have troglomorphic *Astyanax*. Along the southern Sierra de El Abra ridge there are two crests divided by a valley. This is probably linked to a synclinal and two anticlinal folds that comprise the southern El Abra limestone ridge (Figure 11B). We suggest that these ridges and valley may create a divide for cave development that effectively subdivides the hydrologic systems: A western one that drains toward the Río Valles at the town of Ojo de Agua and/or to Río Tampaon at Chiquitita resurgence, and an eastern one that drains to Cueva del Rancho Viejo, Nacimiento del Río Choy, and/or other nacimientos on the eastern escarpment.

In conclusion, we suggest that the combination of tectonics, altitude difference, ancient canyons, ridges, and valleys has affected the karst development of caves in the central and southern Sierra de El Abra. There are barriers for cave-to-cave and surface-to-cave migration and gene flow. Such barriers have affected whole stygobitic communities, as evidenced by the congruence of phyletic markers in both cavefish and mysid shrimp. For *Astyanax*, this has resulted in a group of caves in the central Sierra de El Abra with few instances of hybridization with surface fish and troglomorphic fish harboring mitochondrial haplotype B. In the southwest, there is a population of mixed troglomorphic, epigeomorphic and assumed hybrids which harbor haplotype A, and to the east there are caves with strictly epigeomorphic fish which also harbor haplotype A (Figure 11C). Finally, the central Sierra de El Abra is isolated from the northernmost Pachón cave which harbors the same A mitochondrial haplotype as the southernmost caves studied here. Mitchell et al. [6] proposed a drainage divide, in which the northern Sierra de El Abra drained to Nacimiento del Río Mante and the central and southern portions drained to Nacimiento del Río Choy. The exact location of this barrier is close to Sótano del Venadito, which as of today remains to be genotyped.

### 3.2. Central Sierra de El Abra

#### 3.2.1. Background on Sótano de la Tinaja

A general rule has been invoked to understand the distribution and barriers for dispersal of cavefish [6]. Mainly, that downstream barriers may not be effective to prevent dispersal of cavefish of fish or their eggs because they can simply be flushed down waterfalls. On the contrary, upstream waterfalls may act as effective barriers. Thus, fish may be absent from pools in high passages but not from the lower passages. Based on previous descriptions [6,11], there is one conspicuous exception. Sótano de la Tinaja is described as having cavefish in many pools in the upper levels of the cave, but not below the "Downstream Canyon". This canyon descends steeply from −31 to −82 m depth in a series of four pits, the longest being 18 m. This gallery reaches a large lake room 107 m long, 9 m wide, and 30 m high. The surface of this lake marks the deepest surveyed point in the cave. This lake has yielded a stygobitic shrimp, *Troglomexicanus perezfarfantae* Villalobos, 1974. Curiously, no cavefish have been sighted here. This lake appears to be the major collecting point for flood waters. With the exception of the lake at the cave's deepest point, cavefish have been encountered in many pools and lakes in all the major passages" [11]. It is surprising that biologists visiting the caves in the 1970's did not report the presence of cavefish in the lower levels [11], since there is apparently no barrier for the fish to flow downstream. Therefore, we carried out a new expedition to explore this lake.

#### 3.2.2. Results from Sótano de la Tinaja

Contrary to previous reports [11], *Astyanax* cavefish were abundant in this lake. While no precise method was used for estimating density of fish, a review of underwater videoclips taken in the field showed at least one cavefish for about every 5 m$^2$. The Tinaja lake has a surface area of about 1000 m$^2$.

We also observed abundant *T. perezfarfantae* shrimp and the mysid shrimp S. *quinterensis* at densities of about one shrimp of either species every 5 m$^2$ (Figure 12A,B,E). Less abundant were *Speocirolana pelaezi* Bolivar & Pieltain, 1950 isopods (Figure 12C). Both in the field and later in the laboratory, where cavefish and shrimp were hosted together alive, it was noticed that cavefish did not respond or initiate feeding behavior when close to the shrimp. Cavefish could even bump with a *T. perezfarfantae* shrimp in their swimming paths and not induce feeding behavior. When *Astyanax* was provided with commercial fish food, they immediately conducted feeding behavior, indicating that it was not lack of hunger that prevented them from feeding on the shrimp. Mysid shrimps have also been seen to cohabitate with *Astyanax* at other cave pools in Caballo Moro, Pachón, Sabinos, and the Toro cave system.

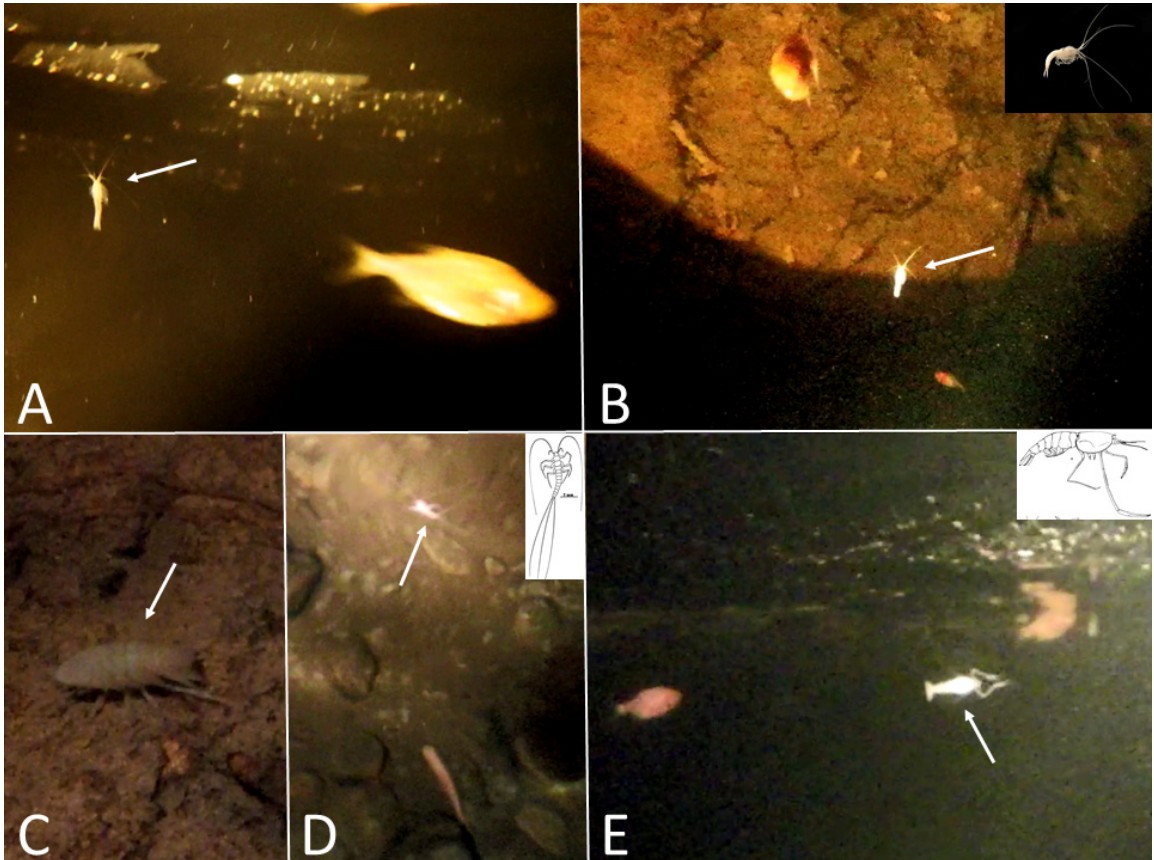

**Figure 12.** Stygobites and troglobites that cohabitate with *Astyanax*, but nonetheless were not seen to actively stimulate feeding behavior on the cavefish despite swimming less than 20 cm from them. (**A**,**B**); Mysid shrimp *Spelaeomysis quinterensis*. (**C**); Isopod *Speocirolana pelaezi*. (**D**); Nicoletiid insect *Anelpistina quinterensis* using surface tension to walk above water. €; Decapod shrimp *Troglomexicanus perezfarfantae*. (**A**), Cueva de Los Sabinos. (**B**–**E**), Sótano de la Tinaja. Inserts in (**B**,**D**,**E**) show higher quality images of organisms of the same genus. All photos were taken in the field. Arrows point to the arthropods.

Another troglobite that did not trigger feeding behavior was the terrestrial nicoletiid insect *A. quinterensis*. While normally it is seen in mud banks, on one occasion an individual was filmed on the water using surface tension to walk. Cavefish swimming just about 20 cm under them did not respond (Figure 12D). On the contrary, a surface cricket that jumped into a pool immediately attracted cavefish and was eaten.

### 3.2.3. Discussion: Central Sierra de El Abra

Previous reports [6,11] specifically said that "no cavefish have been sighted" in the deepest lake of Tinaja cave. Our results showed that, on the contrary, this lake has an abundant population of cavefish. Downstream underground waterfalls do not appear to prevent cavefish dispersal since they can simply be carried down waterfalls. Since the previous authors were able to see the shrimps that inhabit this lake and thus were giving proper attention to the organisms inhabiting this lake, it is unclear if they simply just failed to notice the fish, or abundance of fish was much lower when they explored this portion of the cave. Of ecological interest, it was also found that while *Astyanax* cavefish have evolved a host of enhanced senses to find food in the darkness [1,17], potential stygobite and troglobite prey may have also coevolved to reduce detection or being preyed upon. Thus, they live side by side with *Astyanax* in the same pools and lakes. It may be that adult shrimp are not the standard food of cavefish, and thus they can cohabitate in this lake together.

*3.3. Tamasopo Area*

3.3.1. Background on Cueva del Fraile

　　Cueva del Fraile is two km west of the town of Agua Buena, San Luis Potosí, México. It is in the Tamasopo valley, which is isolated upstream from the general El Abra region by the 100 m high waterfall of Tamul. In a review of *Astyanax* biogeography [11], it was reported that in 1991 three cavers from San Luis Potosí, México, explored Cueva del Fraile and that "about 30 to 50 m inside there was a pool up to 1 m deep, with 15 or 20 pale, eyeless fishes. They got a close look at the fishes in the 2 by 6 m pool, which seemed to be the beginning of a stream passage, but their light was flickering and they had to retreat." The report further says that on November, 2017, Juan Cancino Zapata, a speleologist from Río Verde, San Luis Potosí, and a member of Asociación Potosína de Montañismo y Espeleología (APME), searched for the cave at the request of one of the original explorers. He obtained GPS coordinates and photos. While this was the same cave recalled by the original cavers, Juan Cancino Zapata did not see any fish.

3.3.2. Results from Cueva del Fraile

　　This is a complex cave which, to our knowledge, has yet to be mapped. Our exploration showed that Cueva del Fraile is a resurgence, active only during the rainy season (Figure 13A). The cave is about 200 m away and 50 m above a resurgence that flows year-round (Figure 13B). It is possible that both are connected hydrologically, and that Cueva del Fraile is the high-water discharge conduit. Many galleries in Cueva del Fraile show evidence of becoming phreatic, i.e. with water up to the ceiling during the rainy season (Figure 13D). In many sections fresh mud covers walls and ceiling, suggesting that large amounts of water can circulate through its passages, facilitating movements throughout the cave of any stygobitic organisms inhabiting it. Being a resurgence, water flows from the end of the cave, which is at higher altitude, to the entrance.

　　Galleries are typically at least 5 m high and wide. A few meters after the penumbra zone, a left trending gallery bifurcates. This is the shortest gallery. Several permanent pools are crossed, until a sump is encountered. In the right gallery a maze-like section divides again into two main galleries. The left one crosses several pools until reaching an estimated 15 m pit with a lake at its bottom. Exploration stopped here. The other main gallery also has several pools. It ends at a large room about 30 m high that is covered with fresh mud and blocked by a large collapse from which water flows out of cracks in the rainy season.

　　This cave has many pools, some of them quite large (about 15 m long), which remain full of water throughout the year (Figure 13C,D). During the rainy season the pools are connected in a mostly continuous, water filled conduit. Some pools are inhabited by stygobitic isopods similar to those found in the El Abra caves inhabited by *Astyanax* (Figure 13E). Bats are also encountered over many of these pools, possibly providing food for cave-adapted fish. When compared to the El Abra caves, Cueva del Fraile would appear to provide a perfect habitat for cavefish [21,22]. However, despite careful searching in two independent trips, including snorkeling with scuba diving lights in one of the sumps, we did not find cavefish in any of the pools.

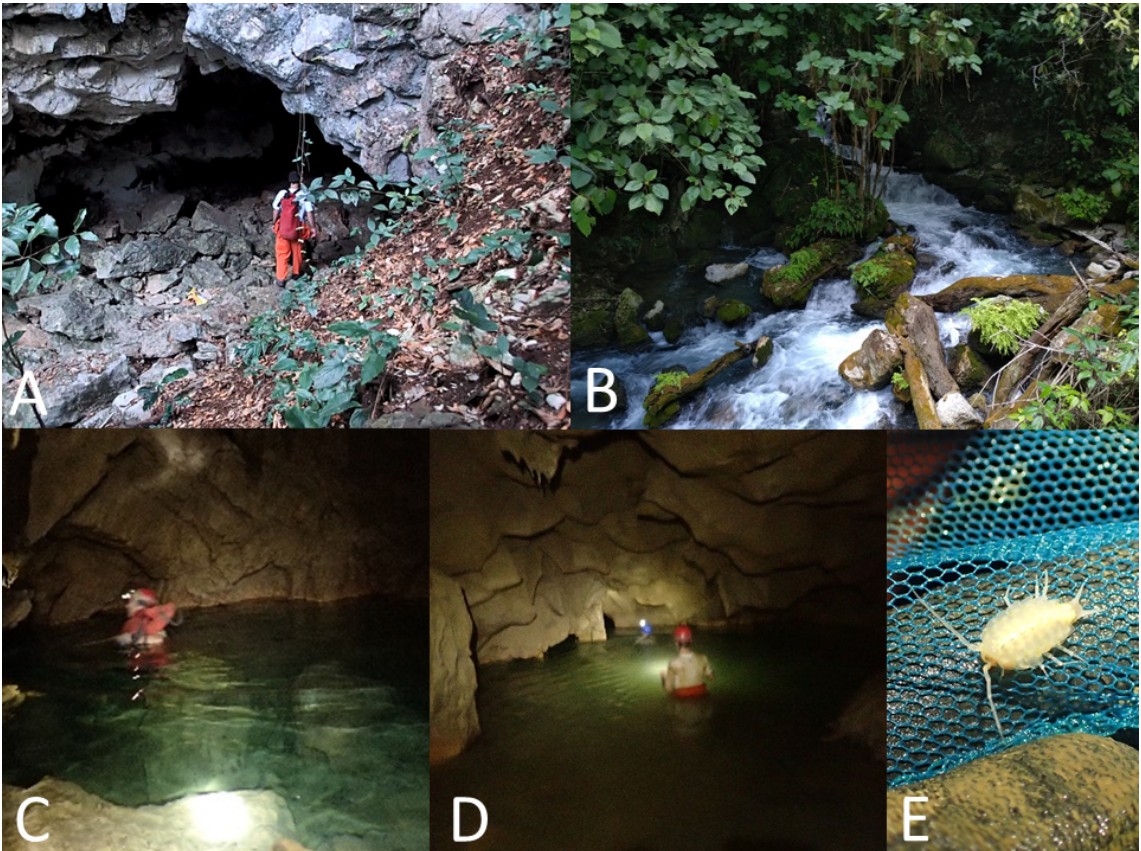

**Figure 13.** Cueva del Fraile in the Tamasopo area. Despite this cave apparently having habitats perfectly suited for *Astyanax* cavefish, none were found. (**A**); Entrance to Fraile cave. (**B**); year-round active resurgence near Cueva del Fraile. (**C,D**); Cueva del Fraile has many pools and lakes. (**E**); Aquatic cave-adapted isopod found in some of these pools. Notice that the roof on D is covered with mud, and shows erosion typical of passages where water reaches the ceiling during the rainy season.

### 3.3.3. Discussion: Tamasopo Area

Due to epigean *Astyanax* showing nyctophilia, i.e. a preference for the darkness [7], surface fish are adept at entering and colonizing the cave environment. In the El Abra region, fully pigmented and eyed fish of presumed surface origin have been found in Cueva de El Mante, Cueva del Rancho Viejo, Cueva Chica del Arroyo Seco, Taninul #2, and Nacimiento del Río Coy ([6,11], and results herein). Additionally, of the 33 described caves with troglomorphic cavefish, 13 have presumed surface fish. Therefore, surface fish are found in at least 18 caves in the El Abra region. Epigean *Astyanax* fish are found throughout the streams of México. Likewise, karstic areas with thousands of caves are found throughout México. This should offer scores of potential sites for the colonization to the underground by *Astyanax*. Nonetheless, if we exclude the two caves in the state of Guerrero which are inhabited by a different species, *A. aeneus* [7,23], all cave adapted *A. mexicanus* ssp. *jordani* populations are found exclusively in the El Abra region. With so many potential places available to establish cave-adapted populations, why is it that only the El Abra area has troglomorphic *A. mexicanus* ssp. *jordani*? It would appear that for some reason, this area had unique and exceptional conditions that allowed for an evolutionary event that culminated in a singular wave or waves of successful adaptation(s) to the underground environment.

The El Abra region is comprised of the Sierra de El Abra and the Sierra de Guatemala, two continuous limestone ranges, plus the adjoining karstic area of Micos, which is only one valley away from the Sierra de El Abra. This close proximity allows for gene flow [9] of adaptive genes

among caves. We assume this is done either by direct underground migration between cave systems as underground drainages are eroded and sequestered, or by local surface populations potentially carrying some cave adapted alleles as "stepping stones".

In an effort to better understand if there was a "singularity" in the El Abra region for *Astyanax* cavefish adaptation, Cueva del Fraile in the Tamasapo valley was visited. Cueva del Fraile had an unconfirmed report of eyeless, depigmented fish [11]. The Tamasopo valley is in a different hydrologic system from the rivers in the El Abra, isolated by the 100 m vertical waterfall of Tamul, which prevents upstream migration from the Río Tampaón. Caves in the Tamasopo valley represent a separate karst area from the Sierra de El Abra, and that hydrologic isolation should have prevented significant flow of adaptive cavernicole genes from El Abra. Therefore, finding a cave-adapted fish population in the Tamasopo valley would indicate a truly independent evolutionary event.

After two failed exhaustive searches for cavefish in Cueva del Fraile, it is our opinion that this cave has no population of eyeless fish. The 1991 report [11] which stated that at about 50 m inside the cave there were eyeless fishes should be considered erroneous. We do not assume that the population went extinct. The report says that although they observed the fish up close, their light was flickering and they had to retreat. Thus, there is the possibility that they actually could not precisely identify the morphs of the fish due to the light limitations, which were already flickering just 50 m away from the entrance. Our assumption is that in 1991 Cueva del Fraile, being a resurgence, probably had surface fish which managed to swim upstream and enter the cave, getting stranded in this pool near the entrance. Many surface fish get pale when kept in the darkness for extended periods. We assume the fish they saw were simply accidental surface fish and not permanent or stable inhabitants of the cave.

Our observations do not support that a troglomorphic population exists deeper in the system. The reason is that barriers for the dispersal of fish tend to be upstream when vertical waterfalls are encountered, not downstream as fish get flushed down waterfalls, especially if passages become phreatic. Cueva del Fraile is a resurgence, and thus at the bottom of the underground hydrologic system. The base of a hydrologic system should be the most accessible part for fish. We propose that with so many suitable large pools available during the dry season in Cueva del Fraile, eyeless fish are not encountered simply because no population of cavefish inhabits this hydrologic system. Our conclusion is supported by the November 2017 report from a speleologist and APME member who searched for the cave at the request of one of the original explorers and also did not see any fish.

Was there a "singularity" in the El Abra region for *Astyanax* cavefish adaptation? While absence of evidence is not evidence of absence, our results further support that the El Abra region experienced exceptional conditions that allowed for a unique evolutionary event. This is supported by decades of studies by cavers and cave biologists throughout México that have failed to find any troglomorphic *A. mexicanus* ssp. *jordani* in caves outside of the El Abra region [6,7,11].

## 4. Conclusions

(1) Two new caves have been found to host cavefish, Sótano del Toro #2 and Sótano de La Calera. These two caves likely interconnect with Sótano del Toro, making up a single hydrologic system.

(2) Previous reports had suggested that Sótano del Toro was inhabited by only troglomorphic fish. Our results show that on the contrary, this system is inhabited by a mixed population of fully eyed and pigmented fish to fully eyeless and depigmented fish.

(3) There is a correspondence in phylogeographical patterns between *Astyanax* cavefish and the stygobitic mysid shrimp *Spelaeomysis quinterensis*. *Astyanax* mitochondrial DNA and mysid histone H3 DNA sequences show that, in both species, cave populations in the central Sierra de El Abra are broadly different from other cave populations. (see also [24]). This phylogeographical convergence supports the notion that the central Sierra de El Abra is a biogeographical zone with effective barriers for either cave-to-cave or surface-to-cave gene flow, which have modulated the evolutionary history of its aquatic community.

(4) The southern portion of the Sierra de El Abra, which includes the Chiquitita, Chica, Cuates, and Toro caves, is a low-lying karst area with apparently few barriers to fish movement, including immigrant surface fish. While undoubtedly there is a lot of underground hydrologic connectivity between many caves [10], there are also biogeographical barriers. Mitchell et al. [6] proposed that there was a hydrological divide somewhere in between Cueva del Pachón and the central Sierra de El Abra caves. Kopp et al. [13] further proposed that there was a biogeographic divide that isolated the central from the southern Sierra de El Abra, which affected both *Astyanax* cavefish and mysid cave shrimps. Our study helped to pinpoint that this divide is somewhere between the Toro and Curva caves. We further propose that there is a third biogeographical barrier to the east. While resurgences on the east side of the El Abra have surface fish and habitats apparently amenable for troglomorphic fish, it appears blind cavefish have not been able to colonize them.

(5) Previous reports had suggested that there were no cavefish in the deepest lake of Tinaja cave. This was an anomaly as downstream barriers are not effective for the dispersal of cavefish since fish (or eggs) can simply be flushed down waterfalls. Our results showed that, contrary to the report, this lake has an abundant population of cavefish.

(6) While *Astyanax* cavefish have evolved a host of adaptive features to help them find food in the darkness, they cohabitate with two stygobitic crustaceans: *T. perezfarfantae* shrimp and the mysid shrimp S. *quinterensis*. Cavefish can be seen swimming barely centimeters away from these crustaceans. Despite being perfectly sized potential prey, they were not seen to activate feeding behavior. Likewise, a troglobitic nicoletiid insect, A. *quinterensis*, was seen walking over water using surface tension without attracting the cavefish. It would appear that while cavefish have evolved to have an enhanced sense of smell [17] and fine-tuned vibration attraction behavior [25,26], among other adaptations to find food in the darkness, potential stygobite and troglobite prey have also coevolved so as to not be detected or be preyed upon.

(7) Report of a cave (Cueva del Fraile) hosting troglomorphic *Astyanax* in the Tamasopo valley, which is outside the general El Abra and Micos contiguous areas, was rejected. Despite decades of cave exploration by cavers and biologist, not a single cave outside the El Abra hydrologic area has been found to host troglomorphic *A. mexicanus* ssp. *jordani*. This supports the concept that the El Abra region experienced exceptional conditions that allowed for a unique evolutionary event resulting in the adaptation to the cave environment by these fish.

**Author Contributions:** Conceptualization, L.E.; field work, L.E., C.P.O.-G., L.L., S.R., R.G.-M., and P.S.; DNA sequencing and morphology; L.E. and A.B.; original draft preparation, L.E. and L.L.; review and editing, all authors. All authors have read and agreed to the published version of the manuscript.

**Funding:** This study was supported by Marist College and its School of Science to LE. An ANR grant [BLINDTEST] and an FRM grant [Equipe FRM] to SR. A collaborative exchange program [Ecos-Nord-CONACYT No. 279100] to SR and POG. Additionally, some field work was supported by UNAM-PAPIIT, project IA203017 of POG. RMG received a Grad student Fellowship at the Posgrado en Ciencia e Ingeniería de Materiales, Universidad Nacional Autónoma de México.

**Acknowledgments:** Thanks to all group members who participated in the field trips: Gerardo Gil, Jessica Gordon, Cait McCann, Matt Schram, Raul Se Feliz, Geraldine Solignac, Juan Cancino Zapata, Víctor Rodríguez Ballesteros, Carlos Pedraza Lara, Morgan Jones, Constance Pierre, Stéphane Père, Julien Fumey, Julien Leclercq, Maryline Blin, Jorge Torres-Paz, Boudjema Imarazene, François Agnès. Students from the BIOL320 Genetics course at Marist College, NY, helped with the DNA sequencing. Proof-reading by Sara Carbone.

**Conflicts of Interest:** The authors declare no conflict of interest. The funders had no role in the design of the study; in the collection, analysis, or interpretation of data; in the writing of the manuscript, or in the decision to publish the results.

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
