# Peer review of "Discovery of Two New Astyanax Cavefish Localities Leads to Further Understanding of the Species Biogeography"

_diversity, doi:10.3390/d12100368_

Round 1
Reviewer 1 Report
General comments:
This study provides important new insights on the distribution and habitats of Astyanax cave morphs. The descriptions of the cave sites are clear, and the conclusions support the observations made during the explorations. There are some minor changes to the text and figures that may improve the manuscript. There are also some discussion points that could be added to incorporate the Northernmost caves.
Throughout the text some of the language sounds informal. For example, starting with the title: “Two new localities of Astyanax cavefish plus revision of its biogeography”. I think “plus” sounds too informal. I think the title could be improved. Perhaps: “Discovery of two new Astyanax cavefish localities leads to further understanding of the species biogeography.”
Introduction and figure 1: Could you refer the reader to the map in the introduction? Please label the different Sierras on the map since you refer to them in the text. Can you include the location of Pachón in one of the maps since you discuss it in the text? Could you also show the locations of the most Northern caves since you refer to them in the methods? It may be important to include a brief description of the Northernmost Molino and Caballo Morro caves with a discussion of how those caves fit into the proposed biogeography. In the methods you say that Molino and Caballo Moro show the A haplotype which is the same as the Southernmost caves. How do you explain this?
Specific comments:
Line 45, delete “-EvoDevo”
Line 53-54: “…may form a barrier.” A barrier to what?
Line 55: Replace “barely” with “nearly” or “approximately”
Line 64-66: Confusing sentence. Maybe change to “Most caves were explored in the 1970s and cavers at this time may have put less effort into describing shallow caves that would nevertheless be of biological interest”
Figure 7. The text needs to be larger.
Figure 8. The text and labels for the caves needs to be larger. Can you center this map over the Toro and Calera caves and also show the distance to Taninul 2 and Rancho Viejo?
Line 290-293: I would argue that they are not “thriving” since they are slimmer. Maybe you could change to: “Although the surface forms appear emaciated compared to cave forms, the large numbers of surface morphs and surface/cave hybrids suggest they can survive and reproduce alongside the cave morph.”
Line 294-297: Mention in the results section that you did not find bats in the Toro caves or provide a reference to this claim.
Line 345: May be easier to understand if re-written: “There is a difference in altitude and a fossil canyon that divides the central and low-lying Southern sections of the Sierra de El Abra”
Line 355 and elsewhere: I believe “East” should be capitalized.
Line 384-386: I found this sentence confusing. Can you re-word of split it in to two sentences?
Line 396: Re-word? “It is unexpected that biologist visiting the caves in the 1970s did not report the presence of cavefish in the lower levels since there is apparently no barrier for the fish to flow downstream. Therefore, we carried out a new expedition to explore this lake.”
Line 429: Did they actually say they checked this pool and no cavefish were found? Or did they not say anything? I am trying to understand if the cavefish are a new population or if you this the previous explorers just didn’t get to this lake.
Line 448: Replace “got” with “obtained”
Line 480: Change to: “…we did not find cavefish in any of the pools.”
Line 483: Delete “Just”
Line 487: Delete “So,” and replace “is” with “are”
Line 489: replace “shows” with “presents”
Line 502: delete “using”
Line 505: I think you should replace “was” with “were”
Line 532: There may be an extra space here.
Author Response
Thank you for your recommendations. They have greatly helped to enhance the quality of the paper. All points were considered and included in the write-up of the new manuscript. In the attached manuscript, the tool “Track changes” was used and it can be seen that all recommendations were included.
Reviewer 2 Report
The authors present an interesting manuscript with qualitative descriptions of several new locations with cave-adapted Astyanax mexicanus present. They speculate that these Tamasopo fish may be a unique evolutionary event from the known events characterized in Dowling et al, 2002.
Further, they describe interconnection with DNA sequencing between several cave-adapted species in phylogeographical patterns, which support that the Sierra de El Abra is a unique zone for cave life.
They also find Tinaja fish in pools previously not seen, supporting the notion of downward barriers not being effective population curbs.
Perhaps most interestingly, they give qualitative descriptions of predator-prey interactions between cavefish and several crustacean species. This may be of profound importance in understanding the nature of cavefish adaptations including feeding, circadian cycles, insulin resistance, and many other traits.
It is of critical importance to the field to gain better understanding of A. mexicanus in their natural context; both for those who use these fish as a lab model system, as well as those who are interested in their evolution and the bio-geography.
The overall quality of writing is average to below average. There are numerous instances with need of significant grammatical editing required. The authors may want to consider finding an editor to assist in addressing many errors. I have pointed out some of these below, but they should carefully read the text and edit as needed.
Comments:
-Line 21: "The eyed morph can also be found in caves, sometimes hybridizing with the cave morph." This sentence is vague in understanding if surface morphs are present in all caves, or just a subset. Please clarify.
-Line 32: "The presumptive location of its boundaries, which may limit cave-to-cave or surface-to-cave gene flow, are identified." It is perhaps too bold of a statement to claim that this manuscript identifies the very complex biological/geographical relationships that constrain cave-cave and surface-cave relationships, especially in regards to gene flow. It would be better to point out that here, you identify new population of A. mexicanus, and that therefore our current knowledge is not complete, but in flux.
-Line 38: The word "great: in the first sentence implies bias and is too colloquial for scientific writing. Use something like "ideal or "optimal"
-Line 39: Should cite the relevant papers
-Line 40: This statement is not widely accepted in the field anymore. If mentioned at all, is should be done with the understanding that while it may previously been widely held that surface and cave morphs are distinct species, that is no longer the prevailing thought. The proper papers supporting this notion should be cited, including the work of Josh Gross, Martina Bradic, Wes Warren, and Suzanne McGaugh. If the authors disagree about the nature of surface and cave fish being 1 or 2 different species, then significant discussion should be added describing each argument without bias. If kept, this sentence also needs significant grammatical editing.
-Line 43: Cite Mcgaugh et al 2014
-Line 49: "Additional" would be better.
-Line 54: I have never seen this species designation written out like this. Please address.
-Line 55: Either give exact dimensions, or say something like "approximately".
-Line 56-57: several english word choices that should be more concise.
-Line 59-60: This is very interesting. Does this knowledge come from gene flow studies, or from actual hydrological studies, or both. Some further mention would be informative for the reader. And if this is the case, have cavefish ever been found in an outside river system? Or is it a 1-way street?
Line 61: I'm not sure if this style of casual discussion of previous literature is common to this field, but generally it is best to simply describe the findings rather than state the authors names or publication years. This sentence would be far better as follows by combining these two sentences: "Previous reviews of caves inhabited by troglomorphic A. mexicanus suggested areas that could yield..."
Line 62: This is far too casual for scientific writing. Please do not use personal pronouns. You can just say "previous studies suggested...."
Line 63: This whole section needs to be re-written.
Line 63b: This sentence feels quite odd here. I think it can be taken out and the following sentences will still work well.
-Line 73: Please format Villalobos 1951 correctly as a citation. It took me a minute to figure out if you were referring to the discovery of the shrimp, or to histone sequencing data.
-Line 80:Please format this correctly as a citation.
-82: Is this published, if so please cite, if not it is speculation, and the language should reflect that, especially since this is of critical importance to your very interesting and significant hypothesis that rightly points out that the evolution of these fish is more complex that we may think.
-Line 86: None of this is needed. Almost all science is published in this format, and adding it here takes away from the great impact of the last sentence of the previous paragraph
-Line 114: delete un-needed word
-Line 145: Combine these two sentences. Please do not describe the cave and pool as small, just list the dimensions that are in the next sentence.
-Line 148: You describe a critical aspect of their environment in such a casual way, and with no scientific value. Do you have any data about how long the sun enters the cave at the time of year you were there? At what intensity the light was in Lux?
-148-150: What is the relevance to this work to include this information?
-Line 150-153: Please fix casual language throughout this section. I am not sure how to directly quote other papers, as it is not something I have seen before, you may want to check with an editor about the proper way to do this.
-Line 162: This should not be in results, even if it is a background section, it is still results, so move to discussion.
-Line 165: Delete un-needed text.
-Line 171: Comparable to surface fish found above ground...
-Line 173: I am confused about this. Why is there a range of fish that were observed? Could you not see them clearly? Can you please state what you actually observed?
-Line 175: Cite Huppop 1987, Espinasa et al 2014, Yoshizawa etal 2010
-Line 181: Delete unnecessary text.
-Line 184: This is confusing. Could you perhaps include a map showing the change in drainage, or further describe. If not, it may be better to take out.
-Line 191: fix grammar
Line 194: Pick a single species name, and cite correctly.
Line 204: unneeded words for results section.
Line 206: Add comma.
Line 214: Please add "A" to figure. This image also looks very stretched, could you fix this? Or perhaps it looks this way due to being a panorama?
Line 219: Unnecessary text for results.
-Line 224-225: No need to compare to human experience. Please simply report based on your findings that this may be a single hydrological system.
-Line 233: Fix casual citation style.
-Line 265: Specify what species these specimens are. It can be a bit confusing reading whether you are referring to fish or shrimp throughout this section.
-Line 280: Can you be more specific? Are these Rascon? Or some other population?
-Line 284: Add text for context
Line 288: clarify with better grammar.
-Line 290-293: This is very interesting. Do you know what the ratio of male-female of the surface fish were that you found? If there are differences, how could this play into maternal effects seen in certain cavefish phenotypes? Would be nice to add a sentence or two about this, citing proper literature. You should also mention here, that surface fish, seem to be highly ammenable to feeding in the dark, citing Yoffe et al 2020 and Lloyd et al 2018.
Line 296: Fix grammar.
-Line 298: Can you speculate why this is? What are the differences in environment?
-Line 302: I realize this is an area of significant debate in the field. But it would probably be worth pointing out that many offer good evidence for the multiple lineage theory, and cite properly, especially Bradic 2012, Coghill 2014, Herman 2018.
-Line 325: fix grammar
-Line 338: fix citation
-Line 381: I would not mention this here, just state that future studies will undertake this task.
-Line 386:Very awkward sentence. Please revise.
-Line 393: grammar, Cite properly.
-Line 407: Cite properly.
Line 408-410: Seems more like discussion than results. Do you have quantitative data showing this feeding behavior? Not necessary, but if you have it would be great.
-Line 429-435: I would move much of what was in the results to this section.
Line 441-449: Please revise, this is far to casual for a scientific publication to fit the style of this journal.
-Line 486: No need for this.
-Line 486-487: It would be really nice if you could make a color-coded figure showing this (which caves have what sort of populations ie pure cavefish, surface fish present, hybrids present). I think it would be a very nice resource for the community showing which caves have what sort of populations.
Line 503: Citation for this?
-Line 514-534: This seems like a very harsh conclusion. I think it would be wise to state, that while it is unlikely based on your findings that there were cavefish, it may be possible that they used to be there. Then defend your position with the evidence you lay out in lines 516-530. It does not destroy your hypothesis if there were cavefish in del Fraile, so since there is no empirical evidence either way, it should not be treated definitively.
-Line 541-542: If these two caves are interconnected hydrologically, should they really be considered two seperate caves?
Line 568-576: This is very interesting. Could you further speculate if these crustaceans may have adaptations that make them unsavory to cavefish?
Author Response
Thank you for your recommendations. They have greatly helped to enhance the quality of the paper. All points were considered and included in the write-up of the new manuscript. In the attached manuscript, the tool “Track changes” was used and it can be seen that all recommendations were included.
We assume that the reviewer had a typo. In his/her introduction it says “they speculate that these Tamasopo fish may be a unique evolutionary event” We are sure the reviewer meant to say “Central Sierra de El Abra” instead of “Tamasopo”. In our paper we show there are no cave-adapted fish in the Tamasopo area.
We did not follow the reviewer’s recommendation regarding changing the format when first presenting a species. The reviewer is not familiar with the Taxonomic Nomenclature Rules for the Binomial Taxonomy, which say that the first time a species name is presented, it should be followed by the author’s name and year. If the species has changed genus affiliation since first described, the author and the year should follow the species name in a parenthesis. As such, when we first introduced a species name in this paper, we included the author and the year. They are part of the species name and thus they are not a citation. The reviewer is incorrect in recommending that we used the [number] with brackets format of a citation. When first introducing a species name, the author, year are not a citation according to the taxonomy nomenclature rules. They are part of the species name. We left as it was Spelaeomysis quinterensis Villalobos, 1951 and other instances when a species was first introduced.